# HSP70-Mediated Autophagy-Apoptosis-Inflammation Network and Neuroprotection Induced by Heat Acclimatization

**DOI:** 10.3390/biology14070774

**Published:** 2025-06-27

**Authors:** Yuchen Su, Xinyan Zheng

**Affiliations:** School of Exercise and Health, Shanghai University of Sport, 200 Hengren Road, Yangpu, Shanghai 200438, China; 22600626@sus.edu.cn

**Keywords:** HSP70, heat acclimatization, autophagy, heat stress, neuroprotection, molecular chaperones

## Abstract

As global warming intensifies the health risks of rising temperatures, scientists are studying how the body protects the brain during heat adaptation. Research reveals that prolonged exposure to high temperatures triggers heat shock protein 70 (HSP70) synthesis, which helps brain cells manage heat stress. This protein coordinates cellular signals to remove damaged components while reducing inflammation and cell death, maintaining brain function. Experiments show its activity varies across brain regions, correlating with heat resistance and potentially reducing heat-related impacts on memory and thinking. These findings suggest new strategies—like drugs or genetic approaches to boost this protein’s effects—to shield the brain from heat damage. Such advances could improve public health protection in warming climates and help people adapt to climate-driven health challenges.

## 1. Introduction

The ongoing and intensifying global climate crisis has resulted in a marked increase in ambient temperatures, thereby giving rise to a series of unprecedented public health challenges associated with thermal stress. A review of the extant epidemiological data reveals a worrying increase in thermoregulatory failures across a severity spectrum, ranging from exertional hyperthermia to life-threatening multiorgan dysfunction [1]. Recent research indicated a near tripling of the global population exposed to extreme heat between 1983 and 2016. This finding was corroborated by a comprehensive study encompassing 13,115 cities. In addition to the phenomenon of climate change, the European region is predicted to witness a 50-fold escalation in heat-related mortality by the year 2100, a trend that is further compounded by the process of increasing urbanization [2]. A comprehensive analysis revealed that individuals afflicted with heat-related illnesses exhibited a 3.18-times elevated risk of developing dementia compared to those who did not [3]. Focusing on the period of 1990–2000, the number of heat-related deaths among people aged 65 years and older increased by 85 per cent, which far exceeded the 38 per cent increase predicted under a stable temperature scenario. In the event of global average temperatures continuing to rise near, but below, the 2 °C threshold, and in the absence of substantial adaptation, it is projected that heat-related deaths will surge by 370% by mid-century, with associated labor losses projected to increase by 50% [4]. This mounting burden underscores the critical absence of current climate-health adaptation frameworks and emphasizes the pressing necessity to investigate protective physiological mechanisms.

During heat stress, the body limits the increase in core temperature by skin vasodilation and sweating [5], a process influenced by the brain and non-thermal signals such as cytokines [6]. Simultaneous vasodilation causes the heart muscle to beat faster and harder, while cardiac filling pressures are reduced [7], thereby increasing cardiovascular burden. Prolonged exposure and heavy sweating can reduce blood volume, and vasodilation can reduce the ability to dissipate heat. This may lead to failure of thermoregulation and significantly increase the likelihood of heat-related illnesses (HRI) [8,9]. Even after body temperature returns to normal, there is an increased risk of cellular damage and an elevated incidence of neurological and cardiovascular diseases [10].

Heat acclimatization (HA) is defined as a complex, multi-system adaptive response to chronic heat exposure, involving physiological, psychological, and structural adaptations [11]. The process is mediated by sensitization of the hypothalamic thermoregulatory center, which coordinates responses such as enhanced sweat secretion, optimized skin blood flow, cardiovascular adjustments, and metabolic modulation [12,13,14]. It has been demonstrated that these adaptations, when considered collectively, enhance the efficiency of heat dissipation while ensuring the maintenance of cardiovascular stability [15]. A fundamental principle of the heat shock response (HSR) is the induction of heat shock proteins (HSPs), with a particular emphasis on the conserved HSP70 family, which functions as molecular chaperones that are indispensable for cellular thermotolerance [16].

The HSP70 family, comprising stress-inducible (HSP72) and constitutive (HSC70) isoforms [17,18], has been demonstrated to exert neuroprotective functions via three primary mechanisms: it plays a central role in maintaining proteostasis by binding to unfolded, misfolded, or aggregated proteins, stabilizing them to prevent denaturation under stressful conditions and facilitating their refolding [19,20]; the suppression of apoptosis through direct inhibition of caspase activation; and the modulation of autophagy flux. A mounting body of evidence underscores the pivotal interactions between HSP70 and cerebral autophagy, a lysosome-dependent degradation pathway that is indispensable for neuronal homeostasis [21]. HSP70 has been shown to regulate autophagy through interconnected pathways, including chaperone-mediated autophagy (CMA), where it is recognized by HSC70 as a substrate and transported through LAMP2A channels [22], and chaperone-assisted selective autophagy (CASA), which involves the BAG3-HSPB8 complex [23].

Notwithstanding the advances that have been made, there are still crucial issues that require resolution. The precise molecular mechanisms by which HSP70 regulates autophagy flux during HA remain to be fully elucidated. Furthermore, the regional specificity of HSP70-induced neuroprotection across brain structures remains to be elucidated. It is indicated that certain neuron-enriched regions, including the cerebellum and the pericortical cell layers, exhibit the induction of HSP70 mRNA following exposure to high temperatures [24]. A more in-depth exploration is necessary to ascertain the therapeutic potential of HSP70 modulation for heat-induced neurological disorders. Examples of this include the prevention and control of heat-induced exacerbation of neurodegenerative diseases (Alzheimer’s disease, Parkinson’s disease) [25], as well as the control of pyrexia [26,27] and malignant hyperthermia [28] through the control of inflammatory responses and the reduction of protein aggregation. This review evaluates the current evidence on the neuroprotective role of HSP70 in HA, focusing on three key areas: (i) the dynamic interactions between HSP70 and cerebral autophagy under thermal stress; (ii) neuroanatomical distribution; and (iii) region-specific protective mechanisms.

## 2. Regulation of HSP70 Expression and Activity During Heat Acclimatization

HSP70 has been identified as a biomarker of cellular heat stress given its acute response to elevated temperature [29]. As a key molecular chaperone, HSP70 stabilizes the cellular proteome by binding to misfolded proteins and aiding their proper refolding [30]. Prolonged exposure to high temperatures reduces the amount of available HSP70, triggering cellular stress responses that activate protective mechanisms, including the synthesis of HSPs [29]. In extreme cases, the failure of these responses can lead to cell death. This thermometer-like role is supported by correlations between HSP70 expression patterns and heat adaptation in animal models [31]. Under normal physiological conditions, HSP70 is also tasked with mediating the correct folding of newly produced proteins as well as their translocation [32].

HSP70 is distributed widely throughout the cytoplasm, nucleus, endoplasmic reticulum, and mitochondria, thus facilitating the maintenance of protein balance [33]. Relative to other organs, the brain has relatively low basal levels. It has been suggested that some mechanism in the brain maintains HSF, which regulates the expression of the HSP70 gene, in a low activity state, limiting the level of HSP70 expression. Furthermore, the internal milieu of the brain is relatively stable due to the presence of the blood–brain barrier, while organs such as the liver and kidneys are in direct contact with the blood and are thus more exposed to external stressors [34]. Aging leads to a substantial decline in the stress-inducing capacity of HSP70, especially its critical ATP-dependent folding function [35], which mainly stems from impaired transcriptional regulation (HSF-1/SIRT1 pathway) [36], resulting in a vulnerable protein homeostatic network that is prone to misfolded protein accumulation and aggregation, promoting age-related diseases such as neurodegeneration. The human exercise-induced HSP70 response exhibits tissue-specific kinetics: acute exercise preferentially triggers rapid upregulation of skeletal muscle mRNA [37,38], whereas prolonged training elevates basal levels and blunts the acute induction capacity [39,40]. This dynamic regulation optimizes cellular stress resistance through the molecular chaperone function of HSP70, supporting muscle adaptation and hypertrophy [41]. HSP70 also has important roles in protecting against cerebral ischemic injury, including inhibition of the apoptotic pathway [28,42] and antagonism of inflammation [27] to significantly reduce infarct size. The process of heat acclimatization has been demonstrated to increase HSP70 levels via a series of physiological responses, including HSF activation, nuclear translocation, and promoter binding. Nevertheless, certain studies have observed discrepancies between the levels of HSP70 protein and mRNA during HA, thereby indicating the presence of a regulatory feedback mechanism [43].

The chaperone activity of HSP70 is contingent upon nucleotide-driven conformational changes. ATP binding to its N-terminal domain has been shown to weaken substrate binding, thereby promoting release [33], while ADP binding has been demonstrated to strengthen substrate attachment [44]. Nucleotide exchange factors (NEFs) have been identified as key regulators of this cycle, functioning to displace ADP from HSP70. This displacement allows ATP binding and subsequent substrate release [45]. Different NEFs have been shown to perform distinct functions. For example, the BAG protein family is a key regulator of the HSP70 cycle, and different BAG proteins can direct release client proteins toward distinct cellular fates. For instance, BAG3 has been observed to reduce protein clumps via HSP70, while BAG1 has been demonstrated to direct proteins for degradation [46,47].

During periods of heat stress, chaperones cooperate in a highly intricate manner. HSP40 facilitates the transfer of unfolded proteins to HSP70 by recognizing hydrophobic regions [48], while HSP90 has been shown to enhance HSP70 production via HSF1 [49]. It has been demonstrated that HA exerts a stimulatory effect on the extracellular levels of HSP70, thereby potentially modulating immune signals [50]. However, the mechanisms by which HSP70 is released extracellularly during HA remain to be elucidated and require further study.

## 3. Autophagy Signaling Pathways Under Heat Stress

Heat stress, defined as physiological and biochemical responses triggered by elevated ambient temperatures, is characterized by protein denaturation, accelerated metabolism, elevated reactive oxygen species (ROS), and heat shock protein (HSP) activation [11]. The aforementioned changes have been shown to disrupt cellular balance, with the activation of autophagy pathways primarily via AMP-activated protein kinase (AMPK) and mechanistic target of rapamycin (mTOR).

AMPK, which is an energy sensor highly conserved throughout evolution, becomes activated when there is an increase in the AMP/ATP ratio [51,52,53]. Under acute heat stress, decreased ATP and increased AMP levels rapidly activate AMPK. This kinase plays a crucial role in cellular metabolism and energy regulation by directly phosphorylating specific residues, such as Ser317 and Ser777, on ULK1 (a crucial kinase complex initiating autophagy), thereby enhancing its activity [13]. It has been demonstrated that phosphorylation at these particular sites on ULK1 further boosts its role and activity in promoting autophagy [54].

The mTOR signaling pathway, which is a central and key regulator of autophagy, exhibits a significant and intricate interaction with AMPK. The mTORC1 complex, a major component of the mTOR pathway, has been shown to act as an inhibitor of autophagy [55]. AMPK has been found to phosphorylate ULK1 at specific sites, namely Thr1227 and Ser1345, which leads to the activation of the tuberous sclerosis complex 2 (TSC2). This, in turn, contributes to the inhibition of mTORC1, as AMPK directly phosphorylates the RAPTOR subunits at Ser722/Ser792 [54]. The inhibition of mTORC1, caused by these molecular interactions, serves to activate ULK1, which then recruits key autophagy-related proteins, such as ATG13, and other essential factors required for initiating the autophagic process [56].

Furthermore, it has been demonstrated that heat-induced protein damage and DNA damage can activate the p53 pathway. Research has consistently shown that the expression of p53 increases as a direct response to heat stress [57]. As a critical tumor suppressor, p53 plays a vital role in cellular responses to a variety of stressors, including DNA damage, by initiating several cellular protective mechanisms. Among these mechanisms, p53 is known to promote autophagy through the activation of targets such as the damage-regulated autophagy modulator (DRAM), a p53-target gene encoding a lysosomal protein that induces macro autophagy [58]. Additionally, p53 has been found to interact with AMPK, which serves to amplify autophagy signals within the cell. AMPK activation promotes autophagy through various mechanisms, including the phosphorylation and activation of the TSC1-TSC2 complex. TSC2 (tuberous sclerosis complex 2) negatively regulates the RHEB G-protein, which positively regulates mTOR. Since mTOR inhibits autophagy, its inhibition by the activated TSC complex (upon AMPK stimulation) enhances autophagy in response to stress conditions. Since mTOR inhibits autophagy, inhibition of the activated TSC complex (under AMPK stimulation) relieves this inhibition and enhances the autophagic response to stress conditions [59].

Heat stress has been demonstrated to disrupt cellular metabolism, resulting in an overproduction of reactive oxygen species (ROS). ROS, along with hypoxia, drives the process of autophagy through two interconnected mechanisms. The first mechanism involves ROS activating the c-Jun N-terminal kinase (JNK) pathway. Once activated, JNK phosphorylates c-Jun, which, in turn, increases the expression of Beclin-1, a critical protein involved in the formation of autophagosomes [60,61]. Beclin-1 is imperative for the initiation of autophagy by promoting the nucleation and expansion of the phagophore, the precursor double-membrane structure that engulfs cytoplasmic cargo to form the autophagosome. The second mechanism is related to hypoxia. Under conditions of low oxygen, the process of stabilization of hypoxia-inducible factor 1 (HIF-1) is initiated, thus allowing it to translocate into the nucleus [62,63]. Upon reaching the nucleus, HIF-1 has been observed to bind to the promoters of BNIP3, a pro-autophagic protein. BNIP3 has been demonstrated to disrupt the interaction between BCL2 and Beclin-1, thereby freeing Beclin-1 to participate in autophagy initiation [62,63,64]. The two mechanisms, driven by ROS and hypoxia, work synergistically to enhance the production of autophagosomes, ultimately promoting the autophagic process in response to cellular stress.

The NF-κB pathway, which is activated by heat-induced inflammatory factors such as tumor necrosis factor alpha (TNF-α), interleukin-1 beta (IL-1β), and reactive oxygen species (ROS) [65]. This activation occurs through the degradation of IκBα, an inhibitory protein that normally prevents the translocation of NF-κB dimers to the nucleus [66,67]. Once IκBα is degraded, NF-κB dimers, such as p65/p50, are released and translocate into the nucleus. In the nucleus, these dimers work to upregulate the expression of autophagy-related genes, including Beclin-1 [68] (Figure 1).

## 4. Expression of HSP70 in Brain Autophagy

Cerebral autophagy, a critical process predominantly occurring in neurons, plays an essential role in maintaining both neuronal development and function. This process is achieved by the efficient clearance of damaged organelles and cellular debris, a prerequisite for the optimal functioning of neurons. This process is of particular importance for maintaining neuronal balance following mitosis [51] and is vital for preserving the structural and functional integrity of neurons during instances of acute injury [69]. However, when autophagy becomes dysregulated, it can lead to detrimental effects, such as the promotion of neuroinflammation [70] or the onset of neurodegenerative conditions [71]. In such cases, the failure to adequately regulate the processes of autophagy contributes to neuronal dysfunction and damage.

HSP70 is one of the key players in controlling the transition from degradation (autophagy) to protein constitution and synthesis at the intracellular level. This transition is mainly achieved through the direct inhibition of autophagy and the strong promotion of protein synthesis and folding. The focus here is on the previously unmentioned function of inhibiting autophagy: HSP70 can interact with the autophagy core protein Beclin-1 [72] to interfere with the binding of Beclin-1 to other pro-autophagic proteins, thereby directly inhibiting autophagy. Additionally, HSP70 helps to maintain protein homeostasis [30] and prevents the accumulation of misfiring proteins. This affects mTORC1 activity and, consequently, autophagy [73].

Heat shock proteins (HSPs) are known to interact with various brain signaling pathways and play a direct role in the regulation of autophagy. Among the autophagy pathways, chaperone-mediated autophagy (CMA) is a selective process that specifically degrades certain proteins through the involvement of the HSP70 protein family. One subtype of the HSP70 family, known as HSC70, binds to the STEM9 motif present in substrate proteins, thereby guiding them to lysosomes for degradation [22]. HSPA8, another member of the HSP70 family, also interacts with KFERQ-like motifs found in target proteins. These interactions facilitate the transport of the proteins to lysosomes via the LAMP2A receptor, further contributing to the selective degradation of proteins through the CMA pathway [22].

HSPs are also involved in chaperone-assisted selective autophagy (CASA), a process that includes the degradation of specific proteins through the mediation of BAG3. HSPB8, a member of the HSP family, has a strong binding affinity for BAG3, which in turn facilitates interactions between HSPB8 and HSPA proteins [74]. This interaction leads to the formation of CASA complexes, composed of HSPA, HSPB8, and BAG3, which collectively work to degrade autophagy-specific target proteins. Although the precise mechanisms linking HSPs to cerebral autophagy are not yet fully understood, emerging research suggests significant connections between these processes. For instance, AMP-activated protein kinase (AMPK), which is activated by changes in the AMP/ATP ratio, has been shown to indirectly enhance the levels of HSPs through an increase in AMP [75]. On the other hand, activation of AMPK has also been observed to reduce the stability of HSP70 mRNA, which leads to a decrease in HSP70 levels [76]. Under conditions of prolonged heat stress, inhibition of AMPK results in the stabilization of HSP70 mRNA, consequently leading to an increase in HSP70 expression [77].

In addition to their involvement in selective autophagy pathways, HSPs also interact with the mTOR pathway, a critical regulator of autophagy and cellular survival. Specifically, HSP90B1, which is a member of the HSP90 family, plays a role in modulating autophagy through the PI3K/AKT/mTOR signaling axis. HSP90 contributes to cell survival, particularly under stress conditions, by ensuring the proper folding of proteins and by inhibiting the apoptotic pathways [78]. In liver cancer models, upregulation of HSP90, HSP70, or HSP27 has been shown to enhance the phosphorylation of AKT and mTOR, leading to a reduction in autophagy while promoting apoptosis. Conversely, inhibiting HSP90 results in a decrease in the levels of these proteins, which, in turn, stimulates both autophagy and apoptosis [79].

The p38 MAPK/mTOR pathway also mediates HSP-driven autophagy. MAPKs, which are stress-activated serine/threonine kinases, are key regulators of the autophagic process. Among these, both p38 MAPK and JNK (c-Jun N-terminal kinase) interact with various HSPs. While the precise function of JNK in HSP-driven autophagy remains to be elucidated, it is understood that JNK exerts its effects through the phosphorylation of c-Jun, which in turn regulates the transcription of HSP genes. Conversely, p38 MAPK activates a range of downstream kinases that are responsible for the phosphorylation of HSP27 [80]. The process of phosphorylation plays a regulatory role in the activation and function of p38 MAPK, thus creating a feedback loop.

The PI3K/Akt pathway crucially regulates HSP70 expression, like MAPK. Both pathways modulate the expression of HSP70 through the action of various transcription factors, such as NF-κB [81,82]. While the PI3K/Akt and MAPK pathways primarily operate independently of one another, they are interconnected and influence the other’s activity [83].

Sirtuin 1 (SIRT1), an NAD^+^-dependent deacetylase [84,85], plays a significant role in regulating the expression of HSPs by deacetylating HSF1. This deacetylation of HSF1 enhances its ability to bind to the promoters of HSP genes [86], thereby promoting their expression. When the activity of SIRT1 is restored, there is an increase in HSP levels, which in turn leads to the suppression of autophagy [32]. The expression of HSPs is intricately linked to the regulation of autophagy, particularly through mechanisms such as CMA and CASA. In this context, the levels of SIRT1 directly influence the acetylation status of HSF1. A reduction in SIRT1 levels results in the increased acetylation of HSF1, which, in turn, impedes its ability to bind to HSP gene promoters, thereby diminishing HSP expression—this phenomenon has been notably observed in neurons [87,88]. HSP25 regulates autophagy by decreasing the interaction between SIRT1 and p53, resulting in the reduced acetylation of p53 [89] and inhibition of the transcriptional activity of p53 [90], which is a core autophagy-regulating gene (Figure 2).

A previous article defined thermotolerance as the ability of a cell or organism to become resistant to heat stress after prior sublethal heat exposure [91]. HSP70 is widely recognized for its essential role in thermotolerance [92] and the regulation of cellular stress responses [93], and it is particularly noted for its neuroprotective functions. Notwithstanding this finding, the protective effects of HSP70 in the brain under conditions of heat stress remain a subject of debate within the scientific community. It has been suggested that there are no significant changes in HSP70 levels in the brain in response to heat stress, in contrast to the positive responses observed in other organs. Conversely, a divergent set of findings has indicated that augmented expression of HSP70 within the brain is associated with a heightened protective effect against heat-induced damage. The observed discrepancies in the literature may be attributed to a variety of factors, including differences in experimental models utilized (for example, in vitro versus in vivo approaches), the specific heat stress conditions applied, or the timing of measurements.

Organ-specific HSP70 responses vary. Mongolian gerbils at 40 °C show elevated liver HSP70 levels but stable brain levels across temperatures (27–43.5 °C) [94]. Similarly, adrenalectomized rats exhibit liver specific HSP70 changes under heat stress, with unchanged brain levels [95], suggesting that the liver prioritizes heat adaptation. This observation was derived from a specific experimental model designed to examine the role of glucocorticoids in organ-specific HSP70 responses. This organ-specific pattern in the absence of glucocorticoids suggests that, under normal conditions, glucocorticoid signaling might be a key factor enabling or modulating the liver’s prioritization of HSP70 induction for heat adaptation, while the brain’s response may be less dependent on this pathway.

Contrastingly, studies highlight HSP70’s brain-protective role. Heat-tolerant goats display higher liver and brain HSP70 levels than sensitive breeds [96]. Broiler chickens show peak HSP70 mRNA levels in the brain [97], and heat-stressed rats due to exercise upregulate central nervous system HSP70, indicating neuronal stress tolerance [98]. Aging studies have found that the process of aging does not affect the brain’s ability to induce HSP70 expression in situations of acute heat stress [99], and that it may be the integrity of molecular chaperone functions that contributes to differences in HSP70 levels. Cerebellar HSP70 activation surpasses hippocampal levels during acute heat stress [100], implying region-specific roles tied to local functions. HSP70 was specifically overexpressed in cerebellar and forebrain fiber tracts under febrile stimulation [101]. High expression of HSP70 in the hippocampus is commonly found in ischemic models [102,103,104]. HSP70 induction also depends on stress duration and intensity.

Neuronal type variability [105] may explain regional brain differences. HSP70 peaks within an hour post-heat stress and normalizes within three hours [106], suggesting rapid, transient protection. This short-term response could clarify conflicting level reports.

HSP70 is linked to cognitive outcomes. Heat-preconditioned rats with high HSP70 levels exhibited improved learning and memory [107]. HSP70 likely aids in injury recovery and survival. Rabbit studies show that oligodendrocyte and microglial HSP70 expression is regulated by MAPK, which also affects hippocampal memory [108], highlighting HSP70’s dual role in survival and neuroplasticity.

In summary, the neuroprotective effects of HSP70 during heat stress are influenced by several factors, including the intensity and duration of exposure, as well as the specific brain region and cell type involved. The enhancement of HSP70 expression has been demonstrated to engender an augmentation in the brain’s resilience to stress and a reduction in damage, the consequence of which is improved cognitive outcomes. This finding indicates that strategies designed to enhance HSP70 levels may offer significant avenues for reducing the impact of heat on health and enhancing brain health under stressful conditions. These findings emphasize the potential therapeutic benefits of modulating HSP70 expression as a means of protecting the brain from the detrimental effects of heat stress.

## 5. HSP70 and Autophagy in HA

Apoptosis is a form of programmed cell death mediated by cascade activation of caspases. This process achieves the removal of damaged or redundant cells by degrading their cellular structures [109,110]. There are three main core pathways: the exogenous pathway triggers caspase-8-dependent apoptosis via cell surface death receptors (e.g., Fas) binding to ligands; the endogenous pathway is regulated by dynamic homeostasis of the Bcl-2 family in the context of cellular stress outer membrane permeabilization (MOMP), which determines mitochondrial apoptosis signaling amplification; and endoplasmic reticulum (ER) stress-induced apoptosis [111].

Autophagy is an evolutionarily conserved catabolic process that degrades abnormal cytoplasmic components via double-membrane autophagosomes wrapping around them and their degradation by lysosomal fusion [112]. Autophagy is activated in response to stress. The mTOR pathway regulates the autophagy switch in response to nutrient/growth factor signaling by phosphorylating the ULK1-ATG13 complex [113,114]. AMPK is a pathway that is activated through the synergistic activation of ULK1 [54], the inhibition of mTORC1 [115], and the regulation of the Beclin-1 complex [116]. AMPK induces autophagy in response to energy stress.

Autophagy and apoptosis form a dynamic bidirectional regulation through key molecules. Autophagy is inhibited by p62-mediated degradation of pro-apoptotic factors (e.g., the BCL-2 family hub molecule NOXA) via ubiquitination to maintain cell survival. Conversely, apoptosis-activated caspases cleave autophagy proteins (e.g., Beclin-1) to terminate autophagy [117,118]. Under certain conditions (e.g., extreme stress), excessive degradation of essential components by autophagy can indirectly induce apoptosis or autophagic death [118]. For example, activation of the transcription factor FOXO3/FOXO3A (fork head box O3), which regulates autophagy when autophagy is defective, leads to apoptosis [119]. Due to the complex crosstalk mechanism between the two, their relationship is still in the exploratory stage. A high heat environment triggers cellular stress, which triggers the initiation of autophagy or apoptosis mechanisms, making heat resistance possible through the dynamic balance of the two mechanisms.

Heat stress activates multiple molecular pathways to trigger autophagy, but its regulation requires both independent pathways and synergistic molecular chaperones. As a core effector of heat acclimatization, HSP70 links heat stress to neuroprotection by mediating autophagic substrate degradation and modulating key signaling nodes. This review examines HSP70’s dual roles in autophagy. Under heat stress, mechanisms like the AMPK/mTOR, p53/DRAM, ROS/JNK/BNIP3, and NF-κB pathways collaborate to regulate autophagy and maintain cellular balance. In heat acclimatization (HA), HSP70 integrates these pathways to bidirectionally control autophagy.

HSP70 has been shown to inhibit the activation of NF-κB [120], thereby reducing the levels of inflammatory cytokines such as TNF-α and IL-1β and indirectly modulating autophagy. This effect occurs because these inflammatory cytokines, in turn, activate NF-κB, which subsequently upregulates autophagy-related genes like Beclin-1 [121] and LC3. By suppressing NF-κB activation, HSP70 may therefore downregulate the expression of these autophagy genes. Moreover, HSP70 interferes with the binding of transcription factors to DNA [122], including NF-κB, which further contributes to a reduction in autophagy-related gene expression. Recent studies have revealed that inhibition of HSP70 enhances the AMPK-mediated phosphorylation of Beclin-1 [72], establishing a direct connection between HSP70 and the regulation of autophagy.

The Bcl-2 family includes anti-apoptotic, pro-apoptotic, and BH3 proteins. Their balance regulates apoptosis via mitochondrial membrane permeability [123]. HSP70 inhibits apoptosis by binding pro-apoptotic Bcl-2 members (e.g., Bax), blocking mitochondrial translocation, and pro-apoptotic factor release [124]. By doing so, HSP70 helps to maintain cellular integrity and prevent premature cell death in response to stress.

HSP70 plays a bidirectional role in regulating autophagy. It not only promotes the formation of autophagosomes through its interaction with Beclin-1 but also counteracts the suppression of autophagy by disrupting the binding of Bcl-2/Bcl-XL to Beclin-1 [125,126]. Beclin-1, along with Vps34, is essential for driving the process of autophagosome formation [127]. In response to stress, acetylated HSP70 binds to the Beclin-1-Vps34 complex, facilitating the assembly of autophagosomes. Furthermore, the activity of Vps34 is enhanced by SUMO proteins, thereby ensuring the efficient and effective formation of autophagosomes [128]. The equilibrium between autophagy and apoptosis highlights the neuroprotective function of HSP70.

HSP70 stabilizes mitochondria, reducing the release of cytochrome c into the cytoplasm and the subsequent activation of caspase 3, which is a key event in the initiation of apoptosis [129]. The release of excess cytochrome c into the cytosol has been shown to inhibit autophagy through the cleavage of Beclin-1, a process that converts Beclin-1 into a pro-apoptotic factor [130]. HSP70 provides protection against apoptosis by blocking death receptor signaling pathways. HSP70 also blocks death receptor signaling by inhibiting Fas transport to the cell surface and caspase-8 activation via dynamin interaction [131,132], further protecting neurons (Figure 3).

Both HSP70 and autophagy are critical for neuroprotection. The exact mechanisms of autophagy in neuroprotection remain unclear, as most studies only confirm its activation after brain injury, with limited exploration of the underlying processes. By contrast, research on HSP70’s neuroprotective roles are more thorough, showing its involvement in CNS inflammation, brain injury, and related conditions. In summary, during HA, HSP70 protects neurons by coordinating core autophagy pathways (e.g., AMPK, mTOR, NF-κB) under heat stress—directly boosting key autophagy proteins (e.g., Beclin-1, Vps34) and indirectly blocking inflammatory and apoptotic signals, achieving molecular-level balance in autophagy.

## 6. Conclusions

During the process of HA, HSP70 functions as a central hub for the maintenance of neuronal homeostasis by means of dynamic regulation of the autophagy-apoptosis-inflammation network. It has been demonstrated through a range of studies that HSP70 can induce protective autophagy (a process by which cells undergo a series of changes to enhance their ability to deal with misfolded proteins and damaged organelles) and promote the clearance of such proteins and organelles in the early stage of heat stress (within hours) by integrating the AMPK/mTOR energy-sensing pathway with the NF-κB inflammatory signaling axis. At the same time, HSP70 has been shown to inhibit the mitochondrial apoptotic pathway (e.g., the imbalance of Bax/Bcl-2) in conjunction with the release of pro-inflammatory factors (TNF-α, IL-1β), functioning as a ‘double insurance policy’ for the purpose of neuroprotection. The inhibition of the mitochondrial apoptotic pathway (e.g., Bax/Bcl2 balance) and the release of pro-inflammatory factors (TNF-α, IL-1β) form a neuroprotective mechanism. It is noteworthy that the spatio-temporal specificity of HSP70 (e.g., high cerebellar expression and rapid response in the hippocampus) suggests that its function is closely related to the metabolic demands of the brain and the history of heat exposure, and that it may be able to form a ‘memory of heat stress’ through epigenetic modifications (e.g., DNA methylation, histone acetylation) to enhance the resistance ability to subsequent heat stress. The findings provide valuable insight into the context of climate warming and the impact of heat exposure on the human body. These findings provide key insights into the molecular basis of neurodegenerative risk in the context of climate warming.

## 7. Future Perspectives

Contemporary research continues to encounter significant challenges. The findings of experiments on rodents are constrained by the disparities in HSP70 expression profiles and blood–brain barrier permeability across species, thereby impeding direct extrapolation to clinical settings. It is recommended that future research efforts concentrate on the development of precise intervention strategies that target the HSP70–autophagy interface. Such strategies may encompass the design of small molecule compounds or gene editing tools that can enhance the synergistic effects of HSP70 and autophagy core proteins (Beclin-1, LC3-II). In addition, the combination of these strategies with biomimetic nano-delivery systems has the potential to overcome the limitations of the blood–brain barrier, thereby achieving regional specificity in the regulation of HSP70. The integration of single-cell multi-omics and organoid models will facilitate the elucidation of the dynamic regulatory network of HSP70 in key cognition-related brain regions and reveal its causal association with heat adaptation phenotypes (e.g., cognitive resilience, metabolic reprogramming). Moreover, the establishment of a multicenter population cohort, in conjunction with the use of wearable devices to monitor individualized heat exposure dose, has the potential to validate the biomarker value of HSP70 expression levels and heat-related neurological damage. This, in turn, would optimize personalized protection programs based on heat adaptation principles. The integration of cross-scale and interdisciplinary research paradigms has the potential to facilitate the translation of the biological properties of HSP70 into proactive defense strategies against the climate crisis. Furthermore, it has the capacity to provide scientific support for the development of a public health system for a climate-resilient society.

## Figures and Tables

**Figure 1 biology-14-00774-f001:**
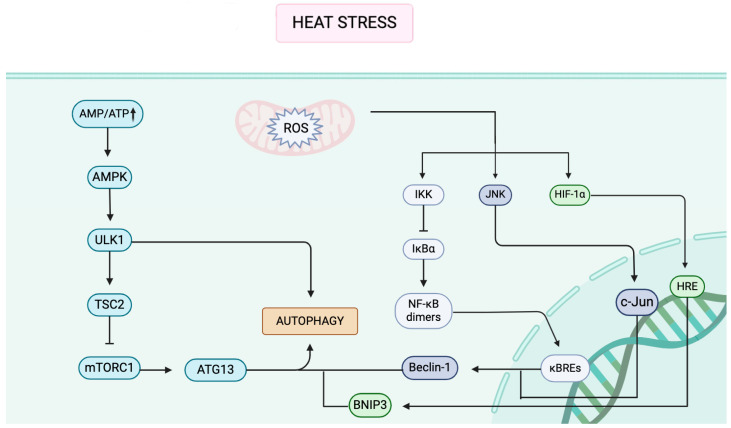
Cellular adaptive responses to heat stress involving energy imbalance, oxidative damage, and autophagy. Heat stress has been demonstrated to trigger cellular adaptive responses through the occurrence of an energy imbalance (AMP/ATP increase, activation of AMPK) and oxidative damage (ROS increase, activation of IKK/NF-κB), activating multiple pathways such as AMPK, NF-κB, and JNK. AMPK, in turn, has been shown to phosphorylate ULK1 and inhibit the TSC2/mTORC1 pathway, thereby promoting autophagy in concert with ATG13. ROS has been shown to regulate inflammatory genes through the degradation of IkBα, which, in turn, releases NF-κB to regulate inflammatory genes. Concurrently, hypoxia mimicry (HIF-1α increase) induces BNIP3, which in turn mediates mitochondrial autophagy (Mitophagy). AMPK: AMP-activated Protein Kinase; NF-κB: Nuclear Factor Kappa-Light-Chain-Enhancer of Activated B Cells; IKK: IκB Kinase; ULK1: Unc-51-Like Autophagy Activating Kinase 1; TSC2: Tuberous Sclerosis Complex 2; mTORC1: Mechanistic Target of Rapamycin Complex 1; ATG13: Autophagy-Related 13; IκBα: Inhibitor of Kappa B Alpha; JNK: c-Jun N-terminal Kinase; c-Jun: v-jun Avian Sarcoma Virus 17 Oncogene Homolog; HRE: Hypoxia Response Element; ↑: Increase.

**Figure 2 biology-14-00774-f002:**
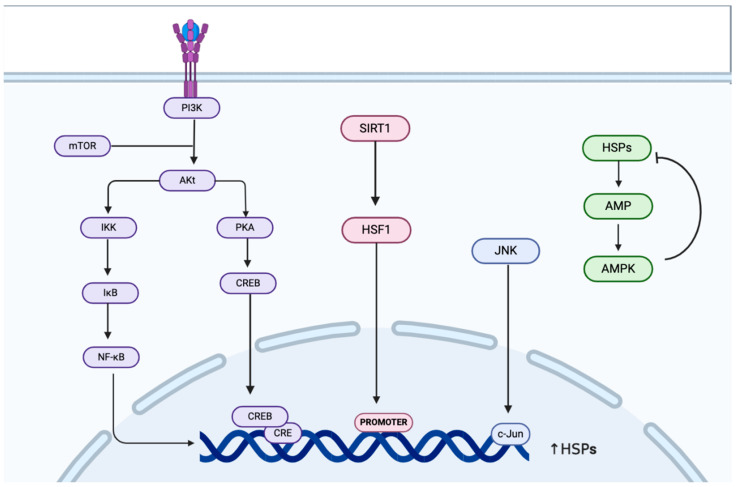
Upregulation of HSPs through the PI3K/AKT/mTOR, AMPK, and SIRT1 signaling pathways. PI3K: Phosphoinositide 3-Kinase; AKT: AK Strain Transforming; mTOR: Mechanistic Target of Rapamycin; PKA: Protein Kinase A; CREB: cAMP Response Element-Binding Protein; CRE: cAMP Response Element; SIRT1: Sirtuin 1; HSF-1: Heat Shock Transcription Factor 1; JNK: c-Jun N-terminal Kinase; c-Jun: v-jun Avian Sarcoma Virus 17 Oncogene Homolog; AMP: Adenosine Monophosphate; AMPK: AMP-activated Protein Kinase.

**Figure 3 biology-14-00774-f003:**
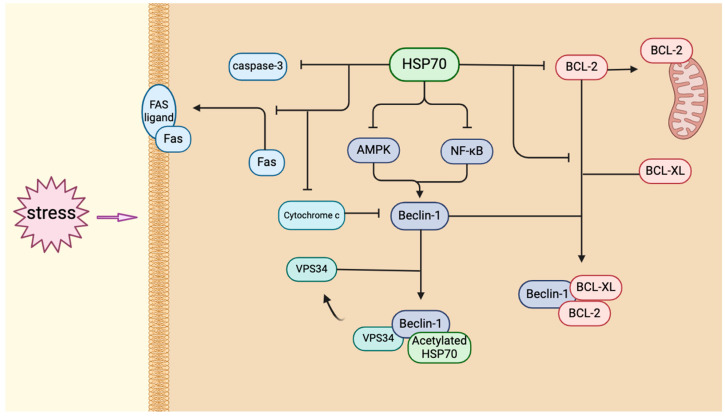
Diverse roles of HSP70 in autophagy modulation and cell survival under stress conditions. HSP70 has been shown to regulate autophagy in order to maintain cellular homeostasis under conditions of cellular stress through multiple mechanisms. It directly promotes the formation of autophagosomes and enhances membrane assembly in concert with Beclin-1 and VPS34; preferentially activates autophagy rather than apoptosis by inhibiting the anti-autophagic effect of BCL-2/BCL-XL or balancing it with cytochrome c; indirectly regulates autophagy by modulating the AMPK and NF-κB stress pathways; and also inhibits caspase-3 activation to provide a time window to promote cell survival. HSP70: Heat Shock Protein 70; AMPK: AMP-activated Protein Kinase; NF-κB: Nuclear Factor Kappa-Light-Chain-Enhancer of Activated B Cells; BCL-2: B-cell Lymphoma 2; Beclin-1: Bcl-2 Interacting Coiled-coil Protein 1; BCL-XL: B-cell Lymphoma Extra Large; VPS34: Vacuolar Protein Sorting-associated Protein 34; Fas: Factor-Associated Suicide.

## Data Availability

No new data were created.

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
