# Peer review of "HSP70-Mediated Autophagy-Apoptosis-Inflammation Network and Neuroprotection Induced by Heat Acclimatization"

_biology, 2025, doi:10.3390/biology14070774_

Round 1
Reviewer 1 Report
Comments and Suggestions for Authors
Major comment. Title “Dynamic Role of HSP70 in Neuroprotection During Heat Acclimatization”
This review examines an important topic related to the role of HSP70 in neuroprotection, as well as the role of autophagy and apoptosis in this process. At the same time, according to the title it should also consider the role of HSP70 in acclimatization of animals to high temperature. In other words, a review with this title should contain an evolutionary comparative analysis of organisms with varying degrees of adaptation to heat stress, with a particular focus on the role of the HSP70 family in the acclimatisation process. This aspect is completely absent from the review, so the title does not correspond to the text.
This review cannot be published in its current form. It is poorly structured. There is no description of the basic concepts, without which it is impossible to understand the individual facts presented.
First, it is necessary to describe the functions of HSP70 under normal conditions and under stress. How does it get activated, and what happens to its activation during ageing? Why are nucleotide exchange factors needed and how do they work? Why is its basal level in the brain low? What happens during mild stress or physical activity? What is thermotolerance and what is its role in protecting against ischaemia?
It is necessary to provide a general description of the processes of apoptosis and autophagy and how they may be related to heat resistance. This information is necessary for understanding the material. The review regularly refers to factors and proteins whose names and functions have not been previously characterised.
Minor comments
- The maintenance of proteostasis by preventing protein denaturation and aiding the refolding of damaged polypeptides - Unclear wording. Where aiding? inadequate reference. Mainly, HSP70 play central role in refolding of misfolded and aggregated proteins. It acts by binding to unfolded or partially folded proteins, helping to stabilize them and prevent them from denaturing under stressful conditions. Can look in reviews doi: 1007/s00018-004-4464-6, https://doi.org/10.1038/nrm2941
- HSP70 has been shown to regulate autophagy through interconnected pathways, including chaperone-mediated autophagy (CMA) via LAMP2A recognition [12] and chaperone-assisted selective autophagy (CASA) involving the BAG3-65 HSPB8 complex [13]. -What it is LAMP2A recognition? how the regulation mechanism works?
- Prolonged exposure to elevated temperatures has been demonstrated to deplete the available levels of HSP70,- thereby triggering cellular stress responses [14]. Extreme case leading to death. Stress response is activation of protective mechanisms, activation of HSP synthesis.
- Nucleotide exchange factors (NEFs) have been identified as key regulators of this cycle, catalyzing the conversion of ADP to ATP [20]. A serious mistake. They do not converse ADP to ATP. Nucleotide exchange factors (NEFs, BAG proteins) eject ADP, allowing ATP rebinding and substrate release.
- As demonstrated in Figure 1, different NEFs have been shown to perform distinct functions. For instance, BAG3 has been observed to reduce protein clumps via HSP70, while BAG1 has been demonstrated to direct proteins for degradation - This is not shown in the figure.
- For instance, BAG3 has been observed to reduce protein clumps via HSP70, while BAG1 has been demonstrated to direct proteins for degradation - Nothing is said about their functions in the previous text.
- ULK1, thereby enhancing its activity. - ULK1: What is this protein? ULK1 is a crucial kinase complex involved in the initiation of autophagy ….
- Furthermore, it has been demonstrated that heat-induced protein and DNA damage can activate the p53 pathway.- What heat induced protein?
- Among these mechanisms, p53 is known to promote autophagy through the activation of targets such as the damage-regulated autophagy modulator (DRAM) –DRAM- function?
- 10. AMPK occurs through the upregulation of both TSC2 and AMPK transcription, further enhancing the autophagic response to stress conditions -TSC2. What is this protein? Function?
- Beclin-1 is imperative for the initiation of the autophagy process by promoting the formation of the autophagy membrane - nothing is said about autophagy membrane.
- Two contradictory statements? explain. 1 Under conditions of heat stress, it has been observed that ATP levels are reduced, while AMP levels increase, leading to the activation of AMPK. 2- Interestingly, under conditions of heat stress, inhibition of AMPK results in the stabilization of HSP70 mRNA, consequently leading to an increase in HSP70 expression [47].
- HSP90 family, plays a role in modulating autophagy through the PI3K/AKT/mTOR signaling axis. HSP90 contributes to cell survival, particularly under stress conditions, by ensuring the proper folding of proteins and by inhibiting the apoptotic pathways. - That's right, but a link is provided on Glucose-regulated protein 94 (GRP94) is an endoplasmic reticulum (ER)-resident member of the heat shock protein 90 (HSP90)”, rather than a general characterisation of the HSP90 family.
- This deacetylation of HSF1 enhances its ability to bind to the promoters of HSP genes- The review does not describe the mechanism of HSF1 action. No references.
- HSP25 is involved in regulating autophagy by modifying the interactions between SIRT1 and p53, ultimately affecting the acetylation status of p53 [56]. - So what? How is this related to HSF deacetylation?
- It has been posited by certain studies that, in the context of heat stress, the brain exhibits comparatively reduced levels of HSP70 in comparison to other organs.- Are you referring to the basal level or induction? If you are comparing the level of HSP70 expression in different tissues, explain why this is the case.
- Similarly, adrenalectomized rats exhibit liver specific HSP70 changes under heat stress, with unchanged brain levels [60], suggesting the liver prioritizes heat adaptation. It is not obvious. - This work is devoted to a special case a possible relationship between the presence or absense of glucocoticoids and Hsp70 content in the liver and brain in male rats during heat stress or LPS-induced fever.
18. Aging studies note consistent HSP70 enrichment across brain regions during heat stress, regardless of age [64]. The paper is not clear on this point. “the present investigation that it is not possible to make a general statement regarding the influence of age on the Hsp70 response in the CNS. In some areas of the brain, age resulted in an attenuation of the heat shock response to hyperthermia, while no such age-related attenuation was present in other areas of the brain “ Also you can read papers https://doi.org/10.3389/fnmol.2023.1230436 , doi: 10.1007/s10522-023-10063-w, DOI: 10.1002/jnr.490270302, https://doi.org/10.1007/978-3-642-76679-4 DOI: 10.1016/j.mad.2004.01.003, PMCID: PMC9277589 , 10.1152/japplphysiol.01267.2001.
Author Response
We appreciate the reviewer’s suggestions, which will help to increase the scientific quality of the paper and improve the resubmitted version. The corrections according to the comments in the revised manuscript are marked in red.
We will answer in a point-to-point fashion to the comments of the reviewer.
Detailed list of changes made to the manuscript, according to comments from Reviewer#1.
Specific feedback
This review examines an important topic related to the role of HSP70 in neuroprotection, as well as the role of autophagy and apoptosis in this process. At the same time, according to the title it should also consider the role of HSP70 in acclimatization of animals to high temperature. In other words, a review with this title should contain an evolutionary comparative analysis of organisms with varying degrees of adaptation to heat stress, with a particular focus on the role of the HSP70 family in the acclimatisation process. This aspect is completely absent from the review, so the title does not correspond to the text.
⇒ Thanks for your comments. We have changed the title to “The role of HSP70 in the protective effect of brain autophagy induced by heat acclimation”. In contrast to the original title this title focuses on the mechanisms rather than the dynamics at different levels of heat stress to ensure that the title is more relevant to the content of the article.
First, it is necessary to describe the functions of HSP70 under normal conditions and under stress. How does it get activated, and what happens to its activation during ageing? Why are nucleotide exchange factors needed and how do they work? Why is its basal level in the brain low? What happens during mild stress or physical activity? What is thermotolerance and what is its role in protecting against ischaemia?
⇒ Thanks for your comments. We have added statements to describe this section in 2. Regulation of HSP70 expression and activity during heat acclimatization:
Original text “This thermometer-like role is supported by correlations between HSP70 expression patterns and heat adaptation in animal models .” The description is of the function of HSP70 under heat stress, and after this sentence we added: “Under normal physiological conditions, Hsp70 is also tasked with mediating the correct folding of newly produced proteins as well as their translocation.” as a supplement to the normal state function of the HSP70.
In the original text, after “HSP70 is distributed widely throughout the cytoplasm, nucleus, endoplasmic reticulum, and mitochondria, thus facilitating the After ‘maintenance of protein balance.”, a paragraph was added to describe the function of HSP70 in low basal levels in the brain, in the aging process, in physical activity, and in the prevention of ischemia: “Relative to other organs, the brain has relatively low basal levels. It has been suggested that some mechanism in the brain maintains HSF, which regulates the expression of the HSP70 gene, in a low activity state, limiting the level of HSP70 expression. Furthermore, the internal milieu of the brain is relatively stable due to the presence of the blood-brain barrier, while organs such as the liver and kidneys are in direct contact with the blood and are thus more exposed to external stressors. Aging leads to a substantial decline in the stress-inducing capacity of HSP70, especially its critical ATP-dependent folding function, which mainly stems from impaired transcriptional regulation (HSF-1/SIRT1 pathway) , resulting in a vulnerable protein homeostatic network that is prone to misfolded protein accumulation and aggregation, promoting age-related diseases such as neurodegeneration. The human exercise-induced HSP70 response exhibits tissue-specific kinetics: acute exercise preferentially triggers rapid up-regulation of skeletal muscle mRNA, whereas prolonged training elevates basal levels and blunts the acute induction capacity, and this dynamic regulation optimizes cellular stress resistance through the molecular chaperone function of HSP70, supporting muscle adaptation and hypertrophy. Hsp70 also has important roles in protecting against cerebral ischaemia injury, including inhibition of the apoptotic pathway, and antagonism of inflammation to significantly reduce infarct size.”
We reproduce the original text “Nucleotide exchange factors (NEFs) have been identified as key regulators of this cycle, catalyzing the conversion of ADP to ATP.” turn into: “Nucleotide exchange factors (NEFs) have been identified as key regulators of this cycle, functioning to displace ADP from HSP70. This displacement allows ATP binding and subsequent substrate release.” Unfolding describes how the NEF works, refining the details of HSP70 activation.
Before “HSP70 is widely recognized for its essential role in thermotolerance” in the original article,we add“A previous article defined thermotolerance as the ability of a cell or organism to become resistant to heat stress after prior sublethal heat exposure.” to explain what is thermotolerance.
It is necessary to provide a general description of the processes of apoptosis and autophagy and how they may be related to heat resistance. This information is necessary for understanding the material. The review regularly refers to factors and proteins whose names and functions have not been previously characterised.
⇒ Thanks for your comments. I have added a description of this section to the original article before “Heat stress activates multiple molecular pathways to trigger autophagy ”: “Apoptosis is a form of programmed cell death mediated by cascade activation of caspases. This process achieves the removal of damaged or redundant cells by degrading their cellular structures. There are three main core pathways: the exogenous pathway triggers Caspase-8-dependent apoptosis via cell surface death receptors (e.g., Fas) binding to ligands; the endogenous pathway is regulated by the dynamic homeostasis of the Bcl-2 family in the context of cellular stress outer membrane permeabilization (MOMP), which determines the mitochondrial apoptotic signaling amplification; and endoplasmic reticulum (ER) stress-induced apoptosis.
Autophagy is an evolutionarily conserved catabolic process that degrades abnormal cytoplasmic components via double membrane autophagosomes wrapped around them and degraded by lysosomal fusion. Autophagy is activated in response to stress. The mTOR pathway regulates the autophagy switch in response to nutrient/growth factor signaling by phosphorylating the ULK1-ATG13 complex. AMPK is a pathway that is activated through the synergistic activation of ULK1, the inhibition of mTORC1 and the regulation of the Beclin-1 complex, AMPK induces autophagy in response to energy stress.
Autophagy and apoptosis form a dynamic bidirectional regulation through key molecules: autophagy is inhibited by p62-mediated degradation of pro-apoptotic factors (e.g., the BCL-2 family hub molecule NOXA) via ubiquitination to maintain cell survival; conversely, apoptosis-activated caspases cleave autophagy proteins (e.g., Beclin-1) to terminate autophagy. Under certain conditions (e.g., extreme stress), excessive degradation of essential components by autophagy can indirectly induce apoptosis or autophagic death. For example, activation of the transcription factor FOXO3/FOXO3A (fork head box O3), which regulates autophagy when autophagy is defective, leads to apoptosis. Due to the complex crosstalk mechanism between the two, their relationship is still in the exploratory stage. The high heat environment triggers cellular stress, which triggers the initiation of autophagy or apoptosis mechanisms, making heat resistance possible through the dynamic balance of the two mechanisms.”
For the factors and proteins not presented in the previous article, they have been further explained according to the comments given in your Minor comments, and you can slide down to see my specific responses to each of your comments.
The maintenance of proteostasis by preventing protein denaturation and aiding the refolding of damaged polypeptides - Unclear wording. Where aiding? inadequate reference. Mainly, HSP70 play central role in refolding of misfolded and aggregated proteins. It acts by binding to unfolded or partially folded proteins, helping to stabilize them and prevent them from denaturing under stressful conditions. Can look in reviews doi: 1007/s00018-004-4464-6, https://doi.org/10.1038/nrm2941
⇒ Thanks for your comments. We have changed it to:“plays a central role in maintaining proteostasis by binding to unfolded, misfolded or aggregated proteins, stabilising them to prevent denaturation in stressful conditions and facilitating their refolding. ” to elucidate how HSP70 maintains protein stability.
HSP70 has been shown to regulate autophagy through interconnected pathways, including chaperone-mediated autophagy (CMA) via LAMP2A recognition [12] and chaperone-assisted selective autophagy (CASA) involving the BAG3-65 HSPB8 complex [13]. -What it is LAMP2A recognition? how the regulation mechanism works?
⇒ Thanks for your comments. We have changed it to:“including chaperone-mediated autophagy (CMA) recognised by HSC70 as a substrate and transported through LAMP2A channels ” Replace inappropriate and unclear expressions in the original text with clear references to the role of LAMP2A in it.
Prolonged exposure to elevated temperatures has been demonstrated to deplete the available levels of HSP70,- thereby triggering cellular stress responses [14]. Extreme case leading to death. Stress response is activation of protective mechanisms, activation of HSP synthesis.
⇒ Thanks for your comments. We have changed it to:“Prolonged exposure to high temperatures reduces the amount of available HSP70, triggering cellular stress responses that activate protective mechanisms, including the synthesis of HSPs. In extreme cases, the failure of these responses can lead to cell death.” Changed to directly correlate the stress response you mentioned with activation of the protective mechanism and point out that hsp synthesis is part of the protective mechanism and added the extreme consequences you mentioned.
Nucleotide exchange factors (NEFs) have been identified as key regulators of this cycle, catalyzing the conversion of ADP to ATP [20]. A serious mistake. They do not converse ADP to ATP. Nucleotide exchange factors (NEFs, BAG proteins) eject ADP, allowing ATP rebinding and substrate release.
⇒ Thanks for your comments. We have changed it to:“ Nucleotide exchange factors (NEFs) have been identified as key regulators of this cycle, functioning to displace ADP from HSP70. This displacement allows ATP binding and subsequent substrate release. ” Replaced my inappropriate descriptions in the original article and revised the original article to the correct theory you presented.
As demonstrated in Figure 1, different NEFs have been shown to perform distinct functions. For instance, BAG3 has been observed to reduce protein clumps via HSP70, while BAG1 has been demonstrated to direct proteins for degradation - This is not shown in the figure.
⇒ Thanks for your comments. We've deleted “As demonstrated in Figure 1”
For instance, BAG3 has been observed to reduce protein clumps via HSP70, while BAG1 has been demonstrated to direct proteins for degradation - Nothing is said about their functions in the previous text.
⇒ Thanks for your comments. “Different NEFs have been shown to perform distinct functions. ”after that I added this sentence::“For example, the BAG protein family is a key regulator of the HSP70 cycle, and different BAG proteins can direct released client proteins towards distinct cellular fates. ” In order to introduce the BAG protein family, naming their main functions.
ULK1, thereby enhancing its activity. - ULK1: What is this protein? ULK1 is a crucial kinase complex involved in the initiation of autophagy ….
⇒ Thanks for your comments. “(a crucial kinase complex initiating autophagy)” was added after ULK1 to briefly describe its function.
Furthermore, it has been demonstrated that heat-induced protein and DNA damage can activate the p53 pathway.- What heat induced protein?
⇒ Thanks for your comments. We replaced “heat-induced protein” with “heat-induced protein damage” to avoid ambiguity.
Among these mechanisms, p53 is known to promote autophagy through the activation of targets such as the damage-regulated autophagy modulator (DRAM) –DRAM- function?
⇒ Thanks for your comments. we add: “a p53-target gene encoding a lysosomal protein that induces macro autophagy and acts as an effector of p53-mediated cell death.” after “Among these mechanisms, p53 is known to promote autophagy through the activation of targets such as the damage-regulated autophagy modulator (DRAM)” to describe what DRAM is.
AMPK occurs through the upregulation of both TSC2 and AMPK transcription, further enhancing the autophagic response to stress conditions -TSC2. What is this protein? Function?
⇒ Thanks for your comments. Inappropriate and unclear statements have been modified: “AMPK activation promotes autophagy through various mechanisms, including the phosphorylation and activation of the TSC1-TSC2 complex. TSC2 (tuberous sclerosis complex 2) negatively regulates the RHEB G-protein, which positively regulates mTOR. Since mTOR inhibits autophagy, its inhibition by the activated TSC complex (upon AMPK stimulation) enhances autophagy in response to stress conditions. Since mTOR inhibits autophagy, inhibition of the activated TSC complex (under AMPK stimulation) relieves this inhibition and enhances autophagic response to stress conditions.” It briefly describes what TSC2 is and elucidates how TSC2 enhances autophagy in response to AMPK stimulation.
Beclin-1 is imperative for the initiation of the autophagy process by promoting the formation of the autophagy membrane - nothing is said about autophagy membrane.
⇒ Thanks for your comments. Do you mean that the term “autophagy membrane” was not explained in the preceding text? As we understand it, the word “autophagy membrane” was replaced with the more rigorous “phagophore” in the original text and further explained. If this is not what we understand, we apologise and would be grateful if you could help us to correct our misunderstanding.
Two contradictory statements? explain. 1 Under conditions of heat stress, it has been observed that ATP levels are reduced, while AMP levels increase, leading to the activation of AMPK. 2- Interestingly, under conditions of heat stress, inhibition of AMPK results in the stabilization of HSP70 mRNA, consequently leading to an increase in HSP70 expression [47].
⇒ Thanks for your comments. “Under conditions of heat stress, it has been observed that ATP levels are reduced, while AMP levels increase, leading to the activation of AMPK.” Here is AMPK activation due to a sharp change in early ATP levels at the minute level. At this time HSP70 levels do not increase due to AMPK activation. And in the second sentence “Interestingly, under conditions of heat stress, inhibition of AMPK results in the stabilization of HSP70 mRNA, consequently leading to an increase in HSP70 expression.” What is shown in the cited article (doi:10.1371/journal.pone.0013096) is an hourly level of change, with ATP levels stabilising (the cited article also states that heat stress for up to 4 hours has no significant effect on intracellular ATP levels), and AMPK levels inhibited, at which point HSP70 expression increases.
What caused this misunderstanding was the time factor, so we made the following changes: replace “Under conditions of heat stress, it has been observed that ATP levels are reduced, while AMP levels increase, leading to the activation of AMPK. ” into “Under acute heat stress, decreased ATP and increased AMP levels rapidly activate AMPK” to highlight the features of acuteï¼›replace “Interestingly, under conditions of heat stress, inhibition of AMPK results in the stabilization of HSP70 mRNA, consequently leading to an increase in HSP70 expression. ” into “Under conditions of prolonged heat stress, inhibition of AMPK results in the stabilization of HSP70 mRNA, consequently leading to an increase in HSP70 expression.”
HSP90 family, plays a role in modulating autophagy through the PI3K/AKT/mTOR signaling axis. HSP90 contributes to cell survival, particularly under stress conditions, by ensuring the proper folding of proteins and by inhibiting the apoptotic pathways. - That's right, but a link is provided on Glucose-regulated protein 94 (GRP94) is an endoplasmic reticulum (ER)-resident member of the heat shock protein 90 (HSP90)”, rather than a general characterisation of the HSP90 family.
⇒ Thanks for your comments. I have replaced the cited article with a review supporting the general characterization of the HSP90 family :exchange“ doi:10.3390/cells10051198”into “DOI: 10.3390/ijms19092560”.
This deacetylation of HSF1 enhances its ability to bind to the promoters of HSP genes- The review does not describe the mechanism of HSF1 action. No references.
⇒ Thanks for your comments. We have added a new citation to support this, this is the link to the article::https://doi.org/10.1016/j.exger.2019.02.011 The abstract section of the article reads:“Mammal sirtuin1 (SIRT1)-mediated deacetylation of heat shock factor 1 (HSF1) upregulates HSF1 binding to the HSP70 promoter.”
HSP25 is involved in regulating autophagy by modifying the interactions between SIRT1 and p53, ultimately affecting the acetylation status of p53 [56]. - So what? How is this related to HSF deacetylation?
⇒ Thanks for your comments. I originally intended to construct the association of p53 with HSP and autophagy here, thanks for pointing out that the logic didn't make sense. I have amended the original text to: “HSP25 regulates autophagy by decreasing the interaction between SIRT1 and p53, resulting in reduced acetylation of p53 and inhibition of the transcriptional activity of p53, which is a core autophagy-regulating gene.”
It has been posited by certain studies that, in the context of heat stress, the brain exhibits comparatively reduced levels of HSP70 in comparison to other organs.- Are you referring to the basal level or induction? If you are comparing the level of HSP70 expression in different tissues, explain why this is the case.
⇒ Thanks for your comments. In the experimental articles I found, many of them made comparisons of HSP70 levels between different organs, and the original intention of writing this sentence was to give a brief summary of the data from the cited articles (doi:10.1016/j.jtherbio.2019.102452,doi:10.1016/S0306-4565(99)00074-1,liver HSP70 levels increased brain HSP70 levels remained unchanged,). After your reminder I realized that there was a discrepancy between the original sentence and what I wanted to convey, so I changed the original sentence to: “It has been suggested that there are no significant changes in HSP70 levels in the brain in response to heat stress, in contrast to the positive responses observed in other organs.”
Similarly, adrenalectomized rats exhibit liver specific HSP70 changes under heat stress, with unchanged brain levels [60], suggesting the liver prioritizes heat adaptation. It is not obvious. - This work is devoted to a special case a possible relationship between the presence or absense of glucocoticoids and Hsp70 content in the liver and brain in male rats during heat stress or LPS-induced fever.
⇒ Thanks for your comments. After “Similarly, adrenalectomized rats exhibit liver specific HSP70 changes under heat stress, with unchanged brain levels, suggesting the liver prioritizes heat adaptation.” , We have added an explanatory paragraph to clarify the specificity of the article: “This observation is derived from a specific experimental model designed to examine the role of glucocorticoids in organ specific Hsp70 responses. This organ-specific pattern in the absence of glucocorticoids suggests that, under normal conditions, glucocorticoid signaling might be a key factor enabling or modulating the liver's prioritization of HSP70 induction for heat adaptation, while the brain's response may be less dependent on this pathway.”
Aging studies note consistent HSP70 enrichment across brain regions during heat stress, regardless of age [64]. The paper is not clear on this point. “the present investigation that it is not possible to make a general statement regarding the influence of age on the Hsp70 response in the CNS. In some areas of the brain, age resulted in an attenuation of the heat shock response to hyperthermia, while no such age-related attenuation was present in other areas of the brain “ Also you can read papers https://doi.org/10.3389/fnmol.2023.1230436 , doi: 10.1007/s10522-023-10063-w, DOI: 10.1002/jnr.490270302, https://doi.org/10.1007/978-3-642-76679-4 DOI: 10.1016/j.mad.2004.01.003, PMCID: PMC9277589 , 10.1152/japplphysiol.01267.2001.
⇒ Thanks for your comments. I carefully read the articles you gave and extracted the part about HSP70 and aging: the first review mentions the correlation between HSP70 levels and aging rate. The results of the second article, which examined changes in the levels of HSP70, HSP40, HSP90, HSF1 DNA binding, and HSF1 protein in the Ames wild type mouse brain and the Ames dwarf type mouse brain, showed that the levels of HSP did not show a consistent pattern of changes to explain the HSF 1- loss of DNA binding activity, and in this level change, the level of HSP70 declined with age.The third article describes the differences in the expression of HSP70 levels in brain regions, mainly in certain neuron-enriched regions, cell layers around the cerebellum and cerebral cortex, and regions of tissue damage showing induction of hsp 70 mRNA. The fourth article uses the rat as an animal model to produce the following results, ‘In the brain, the expression of Hsp72 and Hsc70 increased with age, with the highest levels of induction in the hippocampus and substantia nigra at month 28, followed by the cerebellum, cortex, septum, and striatum.’ The fifth article does not discuss much about neurodegenerative diseases and goes into great detail about the phenomena and mechanisms of normal aging. The sixth article is a very complete introduction to the concept, function and regulation of HSP expression. Thanks again for your recommended articles, I learnt a lot of new arguments and knowledge.
In addition, in conjunction with the article you gave us, we have changed the original text:“Aging studies have found that the process of aging does not affect the brain's ability to induce HSP70 expression in situations of acute heat stress, and that it may be the integrity of molecular chaperone functions that contributes to differences in HSP70 levels. ”

Reviewer 2 Report
Comments and Suggestions for Authors
The manuscript seems interesting, but its current condition does not allow us to recommend its acceptance.
1) The relevance of the study is not obvious. The authors write: "A review of the extant epidemiological data reveals a worrying increase in thermoregulatory failures across a severity spectrum, ranging from exertional hyperthermia to life-threatening multiorgan dysfunction [1]." However, this issue needs to be addressed more fully. It is necessary to provide statistical data, clinical studies and other observations that would confirm the relevance of the review. Perhaps this part could become a separate full-fledged chapter, because it is really interesting (unlike the well-known mechanism of action of heat shock proteins).
2) The authors write "the therapeutic potential of HSP70 modulation for heat-induced neurological disorders", but it is completely unclear what diseases they are talking about and how relevant it is.
The authors write "This review... focusing on ... three key areas: (i) the dynamic
interactions between HSP70 and cerebral autophagy under thermal stress; (ii) the neuro
anatomical distribution and region-specific protective mechanisms; and (iii) the clinical
applications of HSP70 modulation in treating heat-related neurological pathologies." However, the third point is completely ignored in the review.
3) Moreover, the novelty and scientific value of this review is not clear, as the mechanism of action of heat shock proteins is well described in numerous reviews.
Author Response
We appreciate the reviewer’s suggestions, which will help to increase the scientific quality of the paper and improve the resubmitted version. The corrections according to the comments in the revised manuscript are marked in red.
We will answer in a point-to-point fashion to the comments of the reviewer.
Detailed list of changes made to the manuscript, according to comments from Reviewer#2.
Specific feedback
1) The relevance of the study is not obvious. The authors write: "A review of the extant epidemiological data reveals a worrying increase in thermoregulatory failures across a severity spectrum, ranging from exertional hyperthermia to life-threatening multiorgan dysfunction [1]." However, this issue needs to be addressed more fully. It is necessary to provide statistical data, clinical studies and other observations that would confirm the relevance of the review. Perhaps this part could become a separate full-fledged chapter, because it is really interesting (unlike the well-known mechanism of action of heat shock proteins).
⇒ Thanks for your comments. After “A review of the extant epidemiological data reveals a worrying increase in thermoregulatory failures across a severity spectrum, ranging from exertional hyperthermia to life-threatening multiorgan dysfunction [1].” We have added data on heat-related diseases to demonstrate the importance of studies of HA-related mechanisms: “Recent research has indicated a near tripling of the global population exposed to extreme heat between 1983 and 2016. This finding has been corroborated by a comprehensive study encompassing 13,115 cities. In addition to the phenomenon of climate change, the European region is predicted to witness a 50-fold escalation in heat-related mortality by the year 2100, a trend that is further compounded by the process of increasing urbanization. A comprehensive analysis revealed that individuals afflicted with heat-related illnesses exhibited a 3.18 times elevated risk of developing dementia compared to those who did not. Focusing on the period 1990-2000, the number of heat-related deaths among people aged 65 years and older increased by 85 per cent, which far exceeded the 38 per cent increase predicted under a stable temperature scenario. In the event of global average temperatures continuing to rise near, but below, the 2°C threshold, and in the absence of substantial adaptation, it is projected that heat-related deaths will surge by 370% by mid-century, with associated labour losses projected to increase by 50%.”
2) The authors write "the therapeutic potential of HSP70 modulation for heat-induced neurological disorders", but it is completely unclear what diseases they are talking about and how relevant it is.
⇒ Thanks for your comments. After “the therapeutic potential of HSP70 modulation for heat-induced neurological disorders” we add: “Examples of this include the prevention and control of heat-induced exacerbation of neurodegenerative diseases (Alzheimer's disease, Parkinson's disease), as well as the control of pyrexia, and malignant hyperthermia through the control of inflammatory response and reduction of protein aggregation.” To add details of the disease.
The relevance to HSP70 is mentioned later, especially in Part V, which describes how HSP70 regulates autophagy and controls inflammation to maintain neuronal homeostasis under HA. The focus of this paper is to construct the neuroprotection of HSP70 in the thermal environment to demonstrate its therapeutic potential for thermogenic neurological disorders.
The authors write "This review... focusing on ... three key areas: (i) the dynamic
interactions between HSP70 and cerebral autophagy under thermal stress; (ii) the neuro anatomical distribution and region-specific protective mechanisms; and (iii) the clinical applications of HSP70 modulation in treating heat-related neurological pathologies." However, the third point is completely ignored in the review.
⇒ Thanks for your comments. I have removed “and (iii) the clinical applications of HSP70 modulation in treating heat-related neurological pathologies.”. I have only mentioned this point in the Conclusion and future perspectives section at the end, which I have not been able to explore due to the lack of clinical application data, and it is only intended as an expectation and a concept.
3) Moreover, the novelty and scientific value of this review is not clear, as the mechanism of action of heat shock proteins is well described in numerous reviews.
⇒ Thanks for your comments. I am sorry, please allow us to explain the innovative and scientific explanation of this article: our article, in contrast to other reviews that have singularly elaborated the mechanistic basis of action of heat shock proteins, integrates for the first time the molecular mechanistic network of HSP70-autophagy-neuroprotection in heat acclimation, placing HSP70 as the core scheduler of the "autophagy-apoptosis-inflammation " molecular network. It is expected that the principle of the synergistic mechanism of autophagy core protein and HSP70 can provide more new ideas for the treatment and prevention and control of heat-related neurological diseases.

Reviewer 3 Report
Comments and Suggestions for Authors
- In simple summary, Line 11, name or abbreviation of the “protective protein” must be mentioned as just writing protective protein makes it confusing to understand.
- Please change the graphical abstract with high-quality image.
- It would be better to include the “name of specific brain regions” where HSP70’s effects are suspected or have been observed (Line 68-70).
- Rephrase Line 79 for better clarity.
- For all figures; mention the full forms of abbreviations in caption.
- Third focused area of the review as mentioned in Line 75-76 {(iii) the clinical applications of HSP70 modulation in treating heat-related neurological pathologies} should be addressed properly later in the review article. There is no clarity in the manuscript covering this point. Instead, the most discussed thing is “Autophagy” only.
- Some more details covering the “Region-Specific Neuroprotection by HSP70 (High expression in cerebellum and rapid induction in hippocampus)", and "Heat-Induced Neurodegeneration and Protection” should be added to enhance the quality and purpose of this review.
Author Response
We appreciate the reviewer’s suggestions, which will help to increase the scientific quality of the paper and improve the resubmitted version. The corrections according to the comments in the revised manuscript are marked in red.
We will answer in a point-to-point fashion to the comments of the reviewer.
Detailed list of changes made to the manuscript, according to comments from Reviewer#3.
Specific feedback
In simple summary, Line 11, name or abbreviation of the “protective protein” must be mentioned as just writing protective protein makes it confusing to understand.
⇒ Thanks for your comments. We've changed “a protective protein” to “heat shock protein”.
Please change the graphical abstract with high-quality image.
⇒ Thanks for your comments. I'm sorry for the bad reading experience, but the images in the original file we uploaded were of high pixel quality, maybe the images were compressed? If you don't mind, please check if the images are clear in this file we are uploading now.
It would be better to include the “name of specific brain regions” where HSP70’s effects are suspected or have been observed (Line 68-70).
⇒ Thanks for your comments. After “Furthermore, the regional specificity of HSP70-induced neuroprotection across brain structures remains to be elucidated.” we add “It has now been indicated that certain neuron-enriched regions, the cerebellum and the pericortical cell layers, exhibit the induction of HSP70 mRNA following exposure to high temperatures.”
Rephrase Line 79 for better clarity.
⇒ Thanks for your comments. I have changed the original “HSP70 has been described as a molecular "thermometer" due to its rapid response to thermal stress.” to “HSP70 has been identified as a biomarker of cellular heat stress, given its acute response to elevated temperature.” Points out the temperature sensitivity feature mentioned in the cited article and replaces the original uncritical analogy with the more accurate biomarker.
For all figures; mention the full forms of abbreviations in caption.
⇒ Thanks for your comments. We have labelled the full name of the acronym in the comments of the pictures.
Third focused area of the review as mentioned in Line 75-76 {(iii) the clinical applications of HSP70 modulation in treating heat-related neurological pathologies} should be addressed properly later in the review article. There is no clarity in the manuscript covering this point. Instead, the most discussed thing is “Autophagy” only.
⇒ Thanks for your comments. I have removed “and (iii) the clinical applications of HSP70 modulation in treating heat-related neurological pathologies.” I have only mentioned this point in the Conclusion and future perspectives section at the end, which I have not been able to explore due to the lack of clinical application data, and it is only intended as an expectation and a concept.
Some more details covering the “Region-Specific Neuroprotection by HSP70 (High expression in cerebellum and rapid induction in hippocampus)", and "Heat-Induced Neurodegeneration and Protection” should be added to enhance the quality and purpose of this review.
⇒ Thanks for your comments. After “Cerebellar HSP70 activation surpasses hippocampal levels during acute heat stress” we add “HSP70 was specifically overexpressed in cerebellar and forebrain fibre tracts under febrile stimulation. High expression of HSP70 in the hippocampus is commonly found in ischaemic models.”to enrich the details and to give more evidence to support the subsequent conclusions.
After “the therapeutic potential of HSP70 modulation for heat-induced neurological disorders” we add: “Examples of this include the prevention and control of heat-induced exacerbation of neurodegenerative diseases (Alzheimer's disease, Parkinson's disease), as well as the control of pyrexia, and malignant hyperthermia through the control of inflammatory response and reduction of protein aggregation.” The detail that heat induces encephalopathy, including neurodegeneration, is enriched to set the stage and clarify the purpose of the article before expanding on the protective role of HSP70 in this context below.

Reviewer 4 Report
Comments and Suggestions for Authors
Authors Yuchen Su and Xinyan Zheng presented their review of the field of neuroprotective role of HSP70 under heat stress. It is very well known that during elevated temperatures Heat Shock Proteins production is induced. Another well-known fact is neuroprotective features including apoptosis inhibition, protein quality control and participation in different degradation systems. The aim of the following article was apparently to showcase the importance of HSP70 in «afterheat» response. Unfortunately, authors failed to persuasively prove it even though the statement is barely doubtful.
First of all, introduction lacks detailed insight into the physiological outcomes of heat stress and consequences of prolonged exposure.
Throughout the main parts of the text barely any citations are found proving this or that molecular interaction happens after heat stress, for example, line 160, 164, 144, 229-233. Because of that the whole work looks more like the review on the general mechanisms of autophagy regulation by HSP70 rather than its place in these pathways after heat exposure. And also because of that all the thesis made in this review about mechanisms of cellular answer to heat stress and role of HSP70 in it can only be presented as probabilities. I am highly certain that evidence of the statements of this manuscript can be found within a lot of experimental works done in this field. But without that the review does not hold any provable novelty.
Author Response
We appreciate the reviewer’s suggestions, which will help to increase the scientific quality of the paper and improve the resubmitted version. The corrections according to the comments in the revised manuscript are marked in red.
We will answer in a point-to-point fashion to the comments of the reviewer.
Detailed list of changes made to the manuscript, according to comments from Reviewer#4.
Specific feedback
First of all, introduction lacks detailed insight into the physiological outcomes of heat stress and consequences of prolonged exposure.
⇒ Thanks for your comments. We have added a paragraph to the Introduction section to describe the process: “During heat stress the body limits the increase in core temperature by skin vasodilation and sweating, a process influenced by the brain and non-thermal signals such as cytokines. Simultaneous vasodilation causes the heart muscle to beat faster and harder, while cardiac filling pressures are reduced, thereby increasing cardiovascular burden. Prolonged exposure and heavy sweating can reduce blood volume, and vasodilation can reduce the ability to dissipate heat. This may lead to failure of thermoregulation and significantly increase the likelihood of heat-related illnesses (HRI). Even after body temperature returns to normal, there is an increased risk of cellular damage and an elevated incidence of neurological and cardiovascular diseases.”
Throughout the main parts of the text barely any citations are found proving this or that molecular interaction happens after heat stress, for example, line 160, 164, 144, 229-233. Because of that the whole work looks more like the review on the general mechanisms of autophagy regulation by HSP70 rather than its place in these pathways after heat exposure. And also because of that all the thesis made in this review about mechanisms of cellular answer to heat stress and role of HSP70 in it can only be presented as probabilities. I am highly certain that evidence of the statements of this manuscript can be found within a lot of experimental works done in this field. But without that the review does not hold any provable novelty.
⇒ Thanks for your comments. I have added citations to support all the relevant statements for which you have given line numbers, and have additionally added literature support for some of the statements:
160: This activation occurs through the degradation of IκBα, an inhibitory protein that normally prevents the translocation of NF-κB dimers to the nucleus. (PMID: 11134171,PMID: 29862377)
164:In the nucleus, these dimers work to upregulate the expression of autophagy-related genes, including Beclin-1.( PMID: 21994903)
144:In line 144 of the original document,I'm not sure if you are referring to this quote “ROS, along with hypoxia, drives the process of autophagy through two interconnected mechanisms. ” If so, I would like to explain that this sentence is a general introductory sentence to the introduction of the two subsequent mechanisms, and that the relevant literature support is indicated in the introductions of the two subsequent mechanisms.
229-233:The PI3K/Akt pathway crucially regulates HSP70 expression, similar to MAPK. Both of these pathways modulate the expression of HSP70 through the action of various transcription factors, such as NF-κB (PMID: 22005520 ,PMID: 9442376.)While the PI3K/Akt and MAPK pathways primarily operate independently of one another, they are interconnected and influence each other's activity. (PMID: 10576741)
The second mechanism is related to hypoxia. In conditions of low oxygen, the process of stabilization of hypoxia-inducible factor 1 (HIF-1) is initiated, thus allowing it to translocate into the nucleus. (doi:10.1007/s10555-007-9080-0,doi:10.1136/jcp.2003.015032)

Round 2
Reviewer 1 Report
Comments and Suggestions for Authors
Major comments
The revised version of the manuscript entitled "The Role Of HSP70 In The Protective Effect Of Brain Autophagy Induced By Heat Acclimation --" included interest findings. After the general information about the processes discussed was included in the review, the text became easier to understand. However, the new title does not accurately reflect the issues discussed in the review.
When discussing the role of HSP70 in the regulation of autophagy, such an important aspect was not shown that HSP70 can carry out intracellular control, switching the cell from the degradation phase (autophagy) to the phase of protein construction and synthesis.
Minor comment
Research reveals that prolonged exposure to high temperatures triggers heat shock protein that helps brain cells manage heat stress.
Better to say
Research reveals that prolonged exposure to high temperatures triggers heat shock protein 70 (HSP70) synthesis that helps brain cells manage heat stress
Author Response
We appreciate the reviewer’s suggestions, which will help to increase the scientific quality of the paper and improve the resubmitted version. The corrections according to the comments in the revised manuscript are marked in red.
We will answer in a point-to-point fashion to the comments of the reviewer.
Detailed list of changes made to the manuscript, according to comments from Reviewer#1.
Specific feedback
The revised version of the manuscript entitled "The Role Of HSP70 In The Protective Effect Of Brain Autophagy Induced By Heat Acclimation --" included interest findings. After the general information about the processes discussed was included in the review, the text became easier to understand. However, the new title does not accurately reflect the issues discussed in the review.
⇒ Thanks for your comments. We have combined the comments you gave and the summary of the review to rename the article again:“HSP70 mediated autophagy-apoptosis-inflammation network and neuroprotection Induced by Heat Acclimatization”.
When discussing the role of HSP70 in the regulation of autophagy, such an important aspect was not shown that HSP70 can carry out intracellular control, switching the cell from the degradation phase (autophagy) to the phase of protein construction and synthesis.
⇒ Thanks for your comments. We have added a paragraph to the section “4. Expression of HSP70 in brain autophagy “to describe this section: “HSP70 is one of the key players in controlling the transition from degradation (autophagy) to protein constitution and synthesis at an intracellular level. This transition is mainly achieved through the direct inhibition of autophagy and the strong promotion of protein synthesis and folding. The focus here is on the previously unmentioned function of inhibiting autophagy: HSP70 can interact with the autophagy core protein Beclin-1 to interfere with the binding of Beclin-1 to other pro-autophagic proteins, thereby directly inhibiting autophagy. Additionally, HSP70 helps maintain protein homeostasis and prevents the accumulation of misfiring proteins. This affects mTORC1 activity and, consequently, autophagy.”
Research reveals that prolonged exposure to high temperatures triggers heat shock protein that helps brain cells manage heat stress.
Better to say
Research reveals that prolonged exposure to high temperatures triggers heat shock protein 70 (HSP70) synthesis that helps brain cells manage heat stress
⇒ Thanks for your comments. We have replaced the original text with the following:“Research reveals that prolonged exposure to high temperatures triggers heat shock protein 70 (HSP70) synthesis that helps brain cells manage heat stress.”

Reviewer 2 Report
Comments and Suggestions for Authors
The revised manuscript can be accepted.
Author Response
Thank you for your positive decision regarding our manuscript .
Reviewer 3 Report
Comments and Suggestions for Authors
No. All OK.
Author Response

(The authors gave the same response as above.)
